# Reproductive Behavior of the Polyembryonic Parasitoid *Copidosomopsis nacoleiae* (Eady) at Different Ages

**DOI:** 10.3390/insects16030239

**Published:** 2025-02-25

**Authors:** Huili Ouyang, Dongyu Chen, Peng Xiang, Xiaoyun Wang, Wen Lu, Xialin Zheng

**Affiliations:** 1Guangxi Key Laboratory of Agric-Environment and Agric-Products Safety, College of Agriculture, Guangxi University, Nanning 530004, China; 2217404006@st.gxu.edu.cn (H.O.); 15673572572@163.com (P.X.); luwenlwen@163.com (W.L.); zheng-xia-lin@163.com (X.Z.); 2Agricultural Meteorology, Guangxi Institute of Meteorological Sciences, Nanning 530004, China; chendongyu731@163.com; 3National Demonstration Center for Experimental Plant Science Education, College of Agriculture, Guangxi University, Nanning 530004, China

**Keywords:** reproductive behavior, polyembryonic parasitoid, *Copidosomopsis nacoleiae*

## Abstract

In this study, for the first time, we observed and statistically analyzed the reproductive behavior of the polyembryonic parasitoid *Copidosomopsis nacoleiae*, providing a detailed description of its mate-searching, mating, and oviposition behaviors. Additionally, we investigated the life history and field population dynamics of *C. nacoleiae* during its parasitism of *Diaphania angustalis*, with the aim of gaining a deeper understanding of its ecological habits. These results provide a theoretical basis for the field control of *D. angustalis* using *C. nacoleiae*.

## 1. Introduction

Polyembryonic reproduction was first discovered by Marshall (1898) [1]. Through years of entomological research, it has been found in various insect species, primarily in Hymenoptera, although it has also been observed in Hemiptera and Lepidoptera [2,3]. Polyembryonic reproduction is an important reproductive strategy in parasitoid wasps, where a single egg develops into two or more embryos, with each embryo maturing into an individual offspring. Polyembryonic parasitoids typically lay one to eight eggs in a single host. After egg maturation and division, the egg nucleus may undergo one or more divisions, resulting in multiple embryos. In some polyembryonic parasitoid species, competition for nutrients may lead to cannibalism between the embryos, leaving only one embryo to complete development inside the host (polyembryonic uniparasitism), and in some cases, the entire brood may die or suffer partial mortality [4]. Research on the polyembryonic development of parasitoid wasps, especially those that develop endoparasitically within their hosts, remains limited. Notable studies examined the polyembryonic development of *Copidosoma floridanum* (Ashmead) (Hymenoptera: Encyrtidae) in the family Encyrtidae [5]. The embryonic development of this species can be divided into three stages: early cleavage, egg proliferation, and individual morphogenesis. Hu et al. conducted an in-depth study on the polyembryonic development of *Macrocentrus cingulum* (Brischke) (Hymenoptera: Braconidae) in the host *Ostrinia furnacalis* (Guenée) [6], comparing it to the monembryonic development of *Microplitis mediator* (Haliday) (Hymenoptera: Braconidae); they found that the polyembryonic development of *M. cingulum* takes about four times longer than the monembryonic development of *M. mediator* [7]. Hosts parasitized by polyembryonic insects often exhibit noticeably swollen bodies, such as the distorted and desiccated bodies of *Plusia gamma* L. (Lepidoptera: Noctuidae) parasitized by *Litomastix truncatellus* (Dalman) (Hymenoptera: Copidosoma) [8], with the parasitoid wasp’s pupal chambers visible through the deformed irregular body wall. After successful parasitism, hundreds or even thousands of offspring can emerge from a single host. For example, up to 1100 offspring of *C. floridanum* can be produced when parasitizing *Trichoplusia ni* (Hubner) [9].

Polyembryonic parasitoid wasps have a broad range of hosts, including species within the Lepidoptera order, such as the Pyralidae, Olethreutidae, Gracillariidae, Noctuidae, Geometridae, Notodontidae, Bombycidae, Saturniidae, Hesperiidae, and Papilionidae families; the Diptera order, such as the Syrphidae family; the Hemiptera order, including the Aphididae, Pentatomidae, Coreidae, Alydidae, and Fulgoridae families; the Megaloptera order, specifically the Corydalidae family; and the Hymenoptera order, notably the Braconidae family. Key host species include moths, such as *Cnaphalocrocis exigua* (Butler), *Dendrolimus punctatus* (Walker), *Glyphodes vertumnalis* (Guen), *Lamprosema octasema* (Meyrick), *Conopomorpha cramerella* (Snellen), *Acronicta americana* (Harris), *Chartographa fabiolaria* (Oberthür), *Heterocampa guttivitta* (Walker), *Bombyx mori* L., and *Actias luna* Linnaeus, as well as butterflies, such as *Erionota thrax* Linnaeus and *Papilio demoleus* L.

The parasitoid *Copidosomopsis nacoleiae* (Eady) is a polyembryonic endoparasitoid wasp. Its host range includes pests from the Pyralidae, Crambidae, and Olethreutidae families within the Lepidopetra order. These pests include *Cnaphalocrocis patnalis* (Bradley) and *C. ruralis* (Walker), *N. octasema* (Meyr.), *G. vertumnalis*, (Guen), *Lamprosema octasema* (Meyrick), *Parotis vertumnalis* (Guen), and, recently, *D. angustalis* (Snellen) [10,11,12,13,14,15,16,17,18]. *D. angustalis* is one of the most significant defoliators of *Alstonia scholaris* (L.), commonly known as the white cheese tree. Due to its medicinal and ornamental value, *A. scholaris* is widely cultivated in regions of China, such as Guangxi, Guangdong, Hainan, Fujian, Yunnan, and Sichuan. In these areas, the larvae of *D. angustalis* are known to damage the tree [19,20]. The larvae spin silk, causing the leaves of *A. scholaris* to be bundled longitudinally, forming protective “cocoons” where they feed on the leaf tissue, creating a shield around themselves. As a result, chemical control methods are often ineffective against this pest [21]. Studies suggest that *C. nacoleiae* can potentially be used for the biological control of Lepidopteran pests [22]. Recent research on this parasitoid has primarily focused on the morphology of its adult stage and its host range [10,11,12,13]. From 2018 to 2020, our observations in Shangsi County, Fangchenggang City, Guangxi Zhuang Autonomous Region, revealed that the population dynamics of *C. nacoleiae* change in accordance with the occurrence of *D. angustalis*. Except for the months of January to March, the occurrence of both species is proportional throughout the year (unpublished data). However, the reproductive strategies of *C. nacoleiae* parasitic on *D. angustalis* remain unknown, which is of significant importance for scale raising.

Research on various aspects of the reproductive behavior of parasitic wasps, including courtship, mating, oviposition, and host selection, has been conducted both domestically and internationally. The courtship behavior of parasitic wasps follows the usual pattern; wasp males court females by repeatedly tapping the females with their antennae in a series of patterned actions [22,23]. Clear courtship behaviors have been observed in the encyrtid wasp *Ooencyrtus kuvanae* (Howard) [24]. During courtship, the male faces the female. The female holds her antennae downward while the male taps her antennae four times with its forelegs, alternating between the left and right. The female shows no other reaction during the process, appearing stunned by the taps. This sequence defines the courtship behavior of *O. kuvanae*, showing that courtship in egg-parasitic wasps is initiated and led by the male [25]. The male *Psyttalia concolor* (Szépligeti) (Braconidae) exhibits similar behavior, accompanied by rapid wing movements during courtship [26]. Benelli et al. (2012) further investigated the courtship rhythm of *P. concolor*, finding that 1-day-old males courted for an average of 20.3 s. Males courted a single female once or multiple times, with courtship peaking at 9:00 a.m. (8.3 times) and significantly decreasing by 6:00 p.m. [25]. Research on *Aphidius nigripes* (Ashmead) (Braconidae) shows that courtship typically occurs in the morning [27]. Overall, male parasitic wasps court via antennal tapping and wing vibration, with courtship primarily occurring in the morning.

Once a male successfully courts a female, mating begins. The male positions himself behind the female, curving his abdomen forward to mate [26,28]. Meng and Ge (1993) reported that newly emerged *Copidosoma primulus* (Mercet) mate immediately [29]. Damiens and Boivin (2005) found that in *Trichogramma evanescens* (Westwood) (Trichogrammatidae), males have mature sperm upon emergence and are ready to mate [26]. In addition, Benelli et al. (2012) noted that 1-day-old *P. concolor* adults mated for an average of 25.9 s, with peak mating activity at 9:00 a.m. (27.0 instances), which is significantly higher than at 6:00 p.m [30]. After mating, males continue seeking new females or remate with the same female, while females usually refuse additional mating attempts by flying or walking away from the male [28]. Similar behaviors have been observed in the females of *A. nigripes* (Ashmead) and *Aphidius ervi* (Haliday) [27,31]. In summary, parasitic wasps can mate on the day of emergence, with morning mating significantly exceeding afternoon mating. Males tend to mate multiple times, while females typically mate only once. Understanding the mating behavior of parasitic wasps is useful for obtaining more females in artificial rearing and efficiently controlling sex ratios in wasp populations.

The oviposition behavior of female parasitic wasps involves host localization, host inspection and acceptance, oviposition, and grooming [32]. First, females search for hosts in the environment, using their antennae to sense and analyze information. After locating a host, the female conducts a thorough inspection to assess suitability [32]. If the host is deemed suitable, she begins laying eggs. When laying eggs, the heads and tails of *Aprostocetus fukutai* (Miwa and Sonan) (Eulophidae) females arch upwards, while the ovipositor pierces the egg of the *Apriona germari* (Hope) (Cerambycidae) beetle. The body and ovipositor form a “V” shape, which is maintained throughout oviposition [33]. In contrast, during oviposition, the abdomen of *Spalangia endius* (Walker) contracts continuously during oviposition, using the force of contraction to insert the ovipositor vertically into the host. The antennae hover in the air during oviposition and resume tapping when nearing completion [34]. Upon completion, the female withdraws her ovipositor, grooms her antennae, wings, and legs, and departs from the host egg [32,33,34]. Han observed that newly emerged *Telenomus* (Aholcus) *lebedae* Chen et Tong (Scelionidae) females could lay up to 590 eggs in a single day, with a maximum of 12 eggs on a single host. The egg-laying rate decreases with age, accompanied by a reduced parasitism rate [34]. The egg-laying traits of *Anisopteromalus calandrae* (Howard) are similar, with only one egg laid per host, oviposition duration of 0.5–0.6 min, and repeated oviposition behavior [35]. Thus, the oviposition behavior and rhythm of female parasitic wasps are influenced by age, with 1–12 eggs laid per host, decreasing with age [34,35].

Therefore, the developmental process, behavioral patterns, and reproductive rhythms of *C. nacoleiae* parasitic on *D*. *angustalis* were observed and documented to explain their dynamic balance through sophisticated polyembryonic reproductive strategies. These results can not only provide the biological characteristics of *C. nacoleiae* for scale rearing but also gain deeper insights into the evolution, ecological adaptations, and reproductive strategies of parasitic insects.

## 2. Materials and Methods

### 2.1. Insects and Rearing

From 2018 to 2020, the larvae of *D. angustalis*, along with their parasitoid *C. nacoleiae* wasps, were collected monthly from 100 *A. scholaris* trees from Shangsi, Fangchenggang, China (21°44′5″ N, 107°58′54″ E). Five bracts of *A. scholaris* were collected from the upper, middle, and lower layers of the damaged trees in the eastern, southern, western, and northern regions and brought back to the laboratory, where the number of *D. angustalis* larvae in the bracts was counted. Unparasitized 1st- to 4th-instar *D. angustalis* larvae were reared in Petri dishes (20–25 larvae per dish) and fed fresh *A. scholaris* leaves, while 5th- and 6th-instar larvae were transferred to plastic boxes (10–20 larvae per box) and fed mature leaves of the same plant. These leaves, free from pests and diseases, were collected from the vicinity of Guangxi University and the roadsides of Shangsi County. During the mature larval stage, paper trays were provided for pupation. Eggs were laid by the female on the wall of the mating cylinder, and leaves were collected and placed in a Petri dish. Parasitized mature *D. angustalis* larvae were individually housed in Petri dishes with a few fresh leaves. Newly emerged *C. nacoleiae* adults (both male and female) were collected in 10 mL finger tubes. Both *D. angustalis* and *C. nacoleiae* were maintained at 26 ± 2 °C with 70 ± 5% relative humidity and a 14 h light/10 h dark photoperiod.

### 2.2. Life Cycle Analysis of C. nacoleiae

#### 2.2.1. Adults

Emergence Behavior and Circadian Rhythm:

After the host’s color darkened, several pupal chambers of *C. nacoleiae* were dissected from the host body’s surface. The parasitoid pupae were removed and examined using a high-depth imaging system to identify their sex, with records kept. The hosts were then observed daily. Upon the emergence of the wasps, emergence behavior was recorded using an ultra-depth-of-field microscopy system (VHX-6000, Keyence Corporation, Tokyo, Japan). Additionally, 25 host larvae (with identified pupal sex) were each placed in a large Petri dish (D = 20 cm), which was sealed with a film membrane. High-resolution cameras (CS-C1C-1D2WFR, EZVIZ Network Co., Ltd., Hangzhou, China) were used to capture footage of the hosts, and after all the wasps had emerged, videos were reviewed to record the number of *C. nacoleiae* emerging each hour until all the pupae had emerged. The hourly emergence rates of male and female adults were calculated.

Reproductive Characteristics of Adults:

Newly emerged male and female wasps were singularly paired into a 5 mL finger tube with a 10% honey solution and 100 *D. angustalis* eggs. Mating and oviposition behaviors were observed until both male and female wasps died. The pre-mating period, pre-oviposition period, oviposition period, and number of eggs laid by the females were recorded. This experiment was repeated 30 times. Additionally, in a separate experiment with the same conditions, the oviposition behavior of the females was observed until they died. After the experiment, host eggs were dissected, and the number of *C. nacoleiae* eggs in the host eggs was recorded.

Effect of Five Nutritional Conditions on Adult Lifespan:

Newly emerged female wasps were released into five separate rearing cages. Cotton balls with no added substance (blank) or soaked in 5%, 10%, or 20% honey solution or pure water were placed in the cages. Each cage contained 50–80 wasps, and the experiment was repeated three times. Wasp mortality was observed every 24 h, and the time of death and the number of deceased wasps were recorded. The corresponding nutritional source was replaced each time, and the lifespan of the females was determined until all females died. Male wasp lifespans were measured using the same procedure and observation method.

Relationship between Host Characteristics and Number of Emerged Wasps:

Twenty host larvae of *D. angustalis* parasitized by *C. nacoleiae* were sampled and numbered. The body length, width, and weight of these hosts were recorded daily until the wasps emerged. The number of wasps emerging from each host was then recorded. The number of emerging *C. nacoleiae* per host with different body lengths, widths, and weights were compared.

#### 2.2.2. Larvae

Host larvae at the *C. nacoleiae* larval stage were dissected using a high-depth imaging system (VHX-6000, Keyence Corporation, Tokyo, Japan) to observe them, and the wasp larvae within the host were recorded, noting their behavior and developmental habits.

Parasitized *D. angustalis* larvae were dissected using an ultra-depth-of-field microscopy system (VHX-6000, Keyence Corporation, Tokyo, Japan), the number of *C. nacoleiae* wasp larvae were recorded, and physiological and behavioral observations on the pre-imaginal life cycle of the parasitoid were carried out.

#### 2.2.3. Pupae

Hosts at the pupal stage were dissected using a high-depth imaging system to observe and record the parasitoid pupae within the host, and their life habits were documented.

### 2.3. Emergence and Longevity of Adult C. nacoleiae at Various Life Stages

To study the life stages of *C. nacoleiae*, 60 parasitized mature larvae of *D. angustalis* cultured in the laboratory were identified and individually reared. Once the host larvae began to darken, the pupal chambers of *C. nacoleiae* located near the host’s body wall were carefully pierced to extract the parasitoid pupae [36]. The mature pupae were sexually identified using an ultra-depth-of-field microscopy system (VHX-6000, Keyence Corporation, Tokyo, Japan) based on the characteristics of their antennae. The female antennae are nine segments in length, while the male antennae are ten segments in length.

(1)Adult Morphological Characteristics and Lifespan:

On the day of adult emergence, 30 male and 30 female wasps were selected, anesthetized with CO_2_, and photographed to measure their body length and wingspan. Additional newly emerged adults were placed in rearing cages and fed a 10% honey solution. Survival was recorded every 24 h until all individuals died, allowing for the determination of the adult lifespan. Each cage contained 200 adults, with three replicates conducted.

(2)Egg and Larval Morphological Characteristics:

On the day of emergence, 30 pairs of male and female wasps were placed in mating cages (10 cm × 10 cm × 30 cm) containing 300 *D. angustalis* eggs, a 10% honey solution, and fresh leaves of *A. scholaris*, with the date recorded as d1. After 1 day, 100 *D. angustalis* eggs were removed and dissected under a biological microscope to observe the morphology of the parasitoid eggs. Images were taken, egg diameters were measured, and the number of parasitoid eggs per host was recorded. The remaining 200 eggs were placed in Petri dishes to monitor larval development. Upon hatching, daily observations were conducted using a high-depth imaging system to detect signs of parasitism. Dishes or rearing containers were cleaned daily, and fresh *A. scholaris* leaves were provided to feed the *D. angustalis* larvae. The onset of parasitism signs in mature *D. angustalis* larvae marked the larval period of *C. nacoleiae*, at which point individual hosts were identified, labeled, and observed daily. Host dissections were performed with forceps using a high-depth imaging system to examine and measure the larval body length and width of *C. nacoleiae* (N = 30).

(3)Egg–Larval and Pupal Stages: Morphological Characteristics and Duration

When *C. nacoleiae* larvae entered the pupal stage, distinct pupal chambers formed within the host. The date was recorded as d2, and hosts were dissected to extract the pupae, which were photographed and measured for body length and width. Upon adult emergence, the date was recorded as d3. The egg–larval period was calculated as d2–d1 (N = 30), and the pupal period was determined as d3–d2.

### 2.4. Courtship Behavior and Rhythm of Male C. nacoleiae

Males and females of *C. nacoleiae* wasps of the same age (1-day-old females with 1-day-old males or 2-day-old females with 2-day-old males) were paired in a Petri dish (D = 9 cm), each with 10 *D. angustalis* eggs and a 10% honey solution. This was repeated 30 times. Courtship behavior was continuously observed for 24 h using a super-depth three-dimensional microscopy system (VHX-6000, Keyence Corporation, Tokyo, Japan), and high-resolution video (CS-C1C-1D2WFR, EZVIZ Network Co., Ltd., Hangzhou, China) recordings were made. The pairing conditions, replicates, and observation methods for courtship, mating, and oviposition were consistent throughout this study. Courtship behavior was recorded when the male closely followed and chased the female. The start time of courtship (t1) was recorded when the male began to chase the female, and if the male pursued the female for more than 3 s, this was counted as one courtship event [37]. Courtship ended when the male either copulated with the female or abandoned the pursuit. The end time of courtship (t2) was recorded. The number of courtship events per hour and the duration of each courtship event (courtship duration = t2 − t1) were noted.

### 2.5. Mating Behavior and Rhythm of C. nacoleiae Adults

Males and females of *C. nacoleiae* wasps of the same age (1-day-old females with 1-day-old males or 2-day-old females with 2-day-old males) were paired in a Petri dish (D = 9 cm), each with 10 eggs of *D. angustalis* and a 10% honey solution. This experiment was repeated 30 times. Mating behavior was continuously observed for 24 h using a super-depth three-dimensional microscopy system (VHX-6000, Keyence Corporation, Tokyo, Japan), and videos were recorded with a high-definition camera (CS-C1C-1D2WFR, EZVIZ Network Co., Ltd., Hangzhou, China). Mating was considered to have begun when the male grasped the female and extended its abdomen forward to initiate copulation, with the start time (t3) recorded, marking one mating event. The mating was considered complete when the male released the female and retracted its abdomen, and the end time (t4) was recorded. Observations continued until either the female or male died. The number of mating events per hour and the duration of each mating event (mating duration = t4 − t3) were recorded for each *C. nacoleiae* wasp.

### 2.6. Oviposition Behavior and Rhythm of Female C. nacoleiae

Males and females of *C. nacoleiae* wasps of the same age (1-day-old females with 1-day-old males or 2-day-old females with 2-day-old males) were paired in a Petri dish (D = 9 cm), each with 10 eggs of *D. angustalis* and a 10% honey solution. This experiment was repeated 30 times. Oviposition behavior was continuously observed for 24 h using a super-depth three-dimensional microscopy system (VHX-6000, Keyence Corporation, Tokyo, Japan), and videos were recorded with a high-definition camera (CS-C1C-1D2WFR, EZVIZ Network Co., Ltd., Hanzhou, China). The start time of oviposition (t5) was recorded when a mated female resting on the surface of the host egg (*D. angustalis*) secured her body with her legs, retracted her antennae, raised her abdomen, and inserted her ovipositor into the egg, marking one oviposition event. Upon completion, the female rotated her body to withdraw the ovipositor and then left the host egg, at which point the end time of oviposition (t6) was recorded. Observations continued until the female’s death, with the number of oviposition events per hour and the duration of each event (oviposition duration = t6 − t5) recorded for each female. Host eggs were cultured until signs of parasitism emerged, and the parasitism rate was recorded. Additionally, similar oviposition experiments were conducted with mated females under the same pairing conditions, repetitions, and observation methods. After the experiment, parasitized host eggs were dissected under a biological microscope to document the number of eggs laid by each female in each host egg.

### 2.7. Statistical Analysis

Each experiment in this study had at least three biological repetitions. All data are presented as the mean ± standard error (SE). A Kruskal–Wallis H test was used to analyze the effects of supplementary nutrition on the longevity of *C. nacoleiae* adults, the number of host mature larvae with different body weights, the difference in the courtship rate of male *C. nacoleiae* adults at different ages, and the difference in the mating rate of *C. nacoleiae* adults at different ages. All of the curves were *p* = 0.05 in a logarithmic rank test and *p* = 0.00 in a Breslow test. Differences in the number of mature host larvae with different body lengths and body widths and the difference in the oviposition rate of females at different ages were analyzed using Tukey’s statistical method. A Mann–Whitney U test was used to test the number of oviposition events and the duration of oviposition at different ages (in days). The oviposition amount and parasitism rate of different day-old *C. nacoleiae* were tested by a *t*-test.

## 3. Results

### 3.1. Life Cycle of C. nacoleiae

*Copidosomopsis nacoleiae* wasps undergo complete metamorphosis, progressing through four life stages: adult, egg, larva, and pupa. Under laboratory conditions, it takes approximately 48.65 ± 0.49 days (N = 22) to complete one generation. The adult lifespan (with access to 10% honey water) is about 2.12 ± 0.13 days (N = 200, Figure 1); the egg-to-larval stage lasts 32.17 ± 0.20 days (N = 22); and the pupal stage lasts 14.36 ± 0.27 days (N = 22, Figure 1). The eggs, larvae, and pupae of *C. nacoleiae* develop entirely within the host.

The eggs begin to hatch only after the host has matured into a fully developed larva, at which point a significant number of wasp larvae can be observed within the host’s body, indicating parasitism (Figure 1). The *C. nacoleiae* larvae continue their development inside the host before entering the pupal stage. Once the pupae mature, adults emerge from them. After eclosion, the adult *C. nacoleiae* leaves the host to reproduce, employing both sexual reproduction and parthenogenesis. Females of both reproductive modes can lay eggs in the eggs of *D. angustalis* (Figure 1).

### 3.2. Emergence and Longevity of Adult C. nacoleiae of C. nacoleiae

Prior to emergence, *C. nacoleiae* pupae exhibit vigorous movements. As these movements occur, the pupae break through the transparent pupal case, and their heads and appendages begin to move in search of a suitable location to breach the host’s body wall. The wasp uses its mandibles to create an irregular emergence hole in the host’s integument. Once the hole is sufficiently large for the head to protrude, the adult eagerly pushes its way out. Newly emerged adults do not initially fully expand their antennae and wings; they often remain on the surface of the host’s body, grooming their antennae with their forelegs and cleaning their wings with their hind legs. After fully extending their wings, the adults crawl away from the host, completing the emergence process. More than 99.59% of adults emerge between 6:00 and 12:00, with peak emergence for both females and males occurring at 7:00 (Figure 2A). Adults are capable of mating and laying eggs on the same day as their emergence. The pre-mating duration is approximately 4.72 ± 0.24 h (N = 30; same for subsequent measurements), while the pre-oviposition period for females averages 2.80 ± 0.31 h, with an oviposition duration of 4.52 ± 0.12 h (the data reflect both sexual reproduction and parthenogenesis). Females can lay between 2 and 95 clutches throughout their lifetime, averaging 12.75 ± 9.99 clutches, and can produce between 21 and 195 eggs, averaging 107.55 ± 28.38 eggs.

After emergence, *C. nacoleiae* adults require nourishment to enhance their longevity. According to Figure 2B, adults feeding on a 10% sugar solution have the longest lifespan, averaging 2.18 ± 0.09 days (N = 146), significantly different from those on water or 5% sugar solutions (X^2^ = 349.52, *df* = 4, *p* = 0.00, Kruskal–Wallis H test). Adults consuming a 20% sugar solution exhibit some extended lifespans; however, these findings are not significantly different from those for the 10% solution (X^2^ = 349.52, *df* = 4, *p* = 0.00; Figure 2B).

Furthermore, when comparing the effects of different diets on the lifespans of male and female adults, it was found that a 20% sugar solution can extend the lifespans of some males up to 6 days but not those of females (Figure 2C). The impact of the five dietary conditions on lifespans shows significant early differences, with later differences being less pronounced (Figure 2C). In treatments with no food or distilled water, female survival rates are higher than those of males early on, whereas in the 5% to 20% sugar treatments, males show higher early survival rates compared to females (Figure 2C).

The statistical analysis of emergence data indicates that an average of 1489.47 ± 77.54 *C. nacoleiae* individuals can emerge from each host (N = 15), with emergence numbers ranging from 739 to 2024. Figure 3A shows that the emergence rate of *C. nacoleiae* increases with the length of the mature host larvae, peaking when the host measures between 2.90 and 3.10 cm, averaging 1453.56 ± 127.26 individuals, with no significant differences compared to other host size groups (*F* = 0.40, *df* = 2, 44, *p* = 0.67). However, as the width of the mature host larvae increases, the emergence rate first rises and then declines, peaking at 0.48 to 0.52 cm in width, averaging 1380.56 ± 112.47 individuals (*F* = 0.32, *df* = 2, 44, *p* = 0.73; Figure 3B). The emergence rate in relation to the weight of the mature host larvae mirrors that of their width, peaking at weights between 0.24 and 0.33 g, averaging 1433.00 ± 111.51, with no significant differences compared to other weight groups (X^2^ = 1.61, *df* = 2, *p* = 0.45; Figure 3C).

The larvae of *C. nacoleiae* remain active and develop entirely within the host. Shortly after hatching, the larvae begin to move within the host, rapidly spreading throughout its body and consuming its nutrients. They digest all structures within the host, except for the tracheae, gradually taking over the host’s body. At this stage, the host still exhibits vital signs and can move independently. As the larvae grow, they become densely packed inside the host, pressing against one another; eventually, the host dies, and the larvae’s bodies become relatively enlarged (Figure 1B,F).

The pupae of *C. nacoleiae* develop within the host (Figure 1C,G) and have a pupal duration of 14.36 ± 0.27 days. In the early pupal stage, a pupal chamber forms inside the host, characterized by a neat and compact arrangement. Both early and late pupae are inactive, showing no signs of movement. However, late pupae exhibit vigorous movement before emergence, ultimately breaking through the pupal casing to emerge.

### 3.3. Courtship Behavior and Circadian Rhythms of Male C. nacoleiae

The courtship behavior of male *C. nacoleiae* is characterized by the following: Upon locating a target female, the male quickly approaches and uses its antennae to probe the female’s wings, abdomen, back, and legs from behind or the side (Figure 4A). If the female continues to move quickly forward, the male may fail to catch up and may choose to leave, returning to search for another target. Alternatively, if the male confirms the female as a suitable mate through probing, it quickly climbs onto her back, controlling her and initiating copulation, thus concluding the courtship behavior. Observations revealed no courtship behavior from females.

The results of this study on courtship rhythms of males of different ages are illustrated in Figure 4B. One-day-old males become active at 10:00, with courtship rates increasing and peaking at 12:00 with a rate of 44.45 ± 5.92% (N = 30), significantly higher than at any time after 13:00 (*F* = 6.42, *df* = 5, *p* = 0.00, Tukey’s test; the same applies to subsequent comparisons). After 12:00, courtship rates gradually decline, reaching zero by 16:00 (Figure 4C). Two-day-old males achieve a peak courtship rate of 38.89 ± 5.37% at 10:00. Following this peak, the courtship rate gradually decreases, dropping to zero by 14:00. The courtship rate from 10:00 to 11:00 is significantly higher than from 14:00 to 15:00 (X^2^ = 28.96, *df* = 5, *p* = 0.00; Figure 4C). When courtship rates between one-day-old and two-day-old males are compared, the courtship rate of one-day-old males from 14:00 to 15:00 is significantly higher than that of two-day-old males; however, from 10:00 to 11:00, two-day-old males have a significantly higher courtship rate than one-day-old males (Figure 4C).

Observations of courtship behavior revealed that one-day-old and two-day-old males exhibit differences in both the frequency and duration of courtship (Figure 4D,E). The number of courtship attempts by one-day-old males increases starting at 10:00, peaking at 12:00, with an average of 2.67 ± 0.35 attempts (N = 30, Figure 4D), significantly higher than any time after 13:00 (*F* = 6.42, *df* = 5, 179, *p* = 0.00). Two-day-old males peak at 10:00, with an average of 2.33 ± 0.32 attempts (N = 30, Figure 4E), with significantly more attempts from 10:00 to 11:00 compared to 14:00 to 15:00 (X^2^ = 28.96, *df* = 5, *p* = 0.00). In the comparison of courtship attempts between one-day and two-day-old males, one-day-old males show significantly more attempts from 14:00 to 15:00 than two-day-old males (14:00: *U* = −3.09, *df* = 28, *p* = 0.00; 15:00: *U* = −3.08, *df* = 28, *p* = 0.00, Mann–Whitney U test; the same applies to subsequent comparisons). However, from 10:00 to 11:00, two-day-old males exhibit significantly more courtship attempts than one-day-old males (10:00: *U* = −2.91, *df* = 28, *p* = 0.00; 11:00: *U* = −2.90, *df* = 28, *p* = 0.00). Comparing the average daily courtship attempts of males of different ages, one-day-old males (7.50 ± 2.54 attempts) outnumber two-day-old males, but the difference is not significant (t = 0.57, *df* = 28, *p* = 0.58).

The duration of courtship for one-day-old and two-day-old males is shown in Figure 4D,E. One-day-old males peak at a courtship duration of 12.22 ± 3.65 s at 10:00 (N = 30, Figure 4D), although the differences in courtship duration across time periods are not significant (X^2^ = 6.88, *df* = 5, *p* = 0.23). Two-day-old males also reach their peak courtship duration at 10:00, averaging 10.44 ± 2.48 s (N = 30, Figure 4E), which is significantly longer than from 14:00 to 15:00 (X^2^ = 37.85, *df* = 5, *p* = 0.00). When the courtship durations of one-day- and two-day-old males are compared, one-day-old males exhibit significantly longer durations at 12:00 and from 14:00 to 15:00 compared to two-day-old males (12:00: *U* = −2.42, *df* = 28, *p* = 0.01; 14:00: *U* = −3.83, *df* = 28, *p* = 0.00; 15:00: *U* = −3.82, *df* = 28, *p* = 0.00). However, two-day-old males have a longer courtship duration (9.87 ± 0.63 s) than one-day-old males, but the difference is not significant (t = −1.71, *df* = 28, *p* = 0.09).

### 3.4. Mating Behavior and Rhythms of Adult C. nacoleiae

During mating, male wasps use their hind legs and upright wings for support while grasping the female with their forelegs and midlegs, extending their abdomens to mate with the female (Figure 5A). After mating, males release the female and groom their antennae and wings before departing, concluding the mating process. One-day-old adults show increased activity starting at 10:00, with mating rates peaking at 13:00, reaching 21.30 ± 3.91% (N = 30 (same as below), Figure 5B), significantly higher than rates at other times (*F* = 8.77, *df* = 5, 179, *p* = 0.00). After 13:00, the mating rate gradually declines, dropping to zero by 16:00 (Figure 5C). Mating rates of two-day-old adults peak at 10:00, reaching 38.89 ± 5.37% (Figure 5B), after which the rates decrease, with zero mating occurrence by 14:00. The mating rate from 10:00 to 11:00 is significantly higher than from 14:00 to 15:00 (X^2^ = 21.58, *df* = 5, *p* = 0.00, Figure 5C). When the mating rates of one-day-old and two-day-old adults are compared, two-day-old adults exhibit a significantly higher mating rate between 10:00 and 11:00; however, from 13:00 to 15:00, one-day-old adults have a significantly higher rate (Figure 5C).

Research on mating behavior reveals that one-day-old and two-day-old adults differ in both the number and duration of matings (Figure 5D,E). The number of matings for one-day-old adults begins to increase at 11:00, peaking at 13:00 with 1.28 ± 0.23 matings (N = 30, Figure 5D), significantly more than the counts from 10:00 to 11:00 (X^2^ = 23.02, *df* = 5, *p* = 0.00). Two-day-old adults reach a peak of 1.56 ± 0.58 matings at 10:00 (N = 30, Figure 5E), significantly higher than the counts from 14:00 to 15:00 (X^2^ = 18.11, *df* = 5, *p* = 0.00). When the number of matings is compared, one-day-old adults exhibit significantly higher counts than two-day-old adults at both 13:00 and 15:00 (13:00: *U* = −2.35, *df* = 28, *p* = 0.03; 15:00: *U* = −2.68, *df* = 28, *p* = 0.02). Conversely, at 10:00, two-day-old adults have significantly higher mating counts than one-day-old adults (10:00: *U* = −2.68, *df* = 28, *p* = 0.02). In summary, two-day-old adults mate an average of 6.61 ± 1.49 times, significantly more than one-day-old adults (*t* = −4.08, *df* = 28, *p* = 0.00). The duration of mating for one-day-old and two-day-old adults is depicted in Figure 5D,E. In the same environmental conditions, the duration of mating for one-day-old adults initially increases and then decreases, peaking at 13:00 with a duration of 25.11 ± 4.95 s (N = 30, Figure 5D), which is significantly longer than the mating duration between 14:00 and 15:00 (X^2^ = 20.32, *df* = 4, *p* = 0.00). The mating duration for two-day-old adults also shows an initial increase, followed by a decrease, peaking at 12:00 with a value of 13.44 ± 1.71 s (N = 30, Figure 5E), although differences among time periods are not significant (*F* = 2.22, *df* = 3, 123, *p* = 0.11). Comparing mating durations between one-day-old and two-day-old adults reveals that one-day-old adults have longer mating durations at 11:00 and 13:00 (11:00: *t* = 2.60, *df* = 28, *p* = 0.02; 12:00: *t* = −0.12, *df* = 28, *p* = 0.91; 13:00: *t* = 2.34, *df* = 28, *p* = 0.04). Overall, one-day-old adults have a longer average mating duration of 14.20 ± 1.69 s, although the difference compared to two-day-old adults is not significant (*U* = −1.41, *df* = 28, *p* = 0.16).

### 3.5. Oviposition Behavior and Rhythms of Female C. nacoleiae

Under laboratory conditions, female *C. nacoleiae* can lay eggs on the day of emergence. After mating, the female begins to search for suitable hosts to oviposit. Initially, the female first crawls near the eggs of the *D. angustalis* moth, frequently tapping the egg surface with its antennae. Once the female finds an appropriate host, it stops moving, anchors its body with its legs, retracts its antennae, elevates its abdomen, and pierces the eggs of the *D. angustalis* moth with its ovipositor to lay its eggs. Upon completing oviposition, the female twists its body to withdraw the ovipositor and quickly leaves the host’s eggs.

The egg-laying rate of one-day-old females gradually increases starting at 10:00, peaking at 12:00 with a rate of 88.89 ± 5.17% (N = 30, Figure 6A). This rate then declines gradually, reaching zero by 16:00. The egg-laying rate from 10:00 to 12:00 is significantly higher than from 14:00 to 15:00 (*F* = 14.56, *df* = 5, 35, *p* = 0.00, Figure 6A). The peak of the oviposition of two-day-old females occurs at 10:00, with a rate of 16.67 ± 4.06% (N = 30, Figure 6A). The rate then decreases gradually, also reaching zero by 16:00, with the rate at 10:00 being significantly higher than from 12:00 to 15:00 (*F* = 6.10, *df* = 5, 35, *p* = 0.00, Figure 6B).

Comparing the oviposition rates of different-aged females, one-day-old females exhibit significantly higher rates from 10:00 to 15:00 than two-day-old females (Figure 6B). Observations of the oviposition behavior of female *C. nacoleiae* reveal that the times of oviposition of one-day-old females was 69.33 ± 7.10, which was significantly more than two-day-old females (Figure 6C). The duration of oviposition for one-day-old females is 101.79 ± 6.07 s, which is longer than that of two-day-old females, with a significant difference between the two (Figure 6D). Additionally, the fecundity of one-day-old females (8.84 ± 0.43 eggs) was significantly higher than that of two-day-old females (Figure 6E). However, when the parasitism rates of different-aged females are compared, the rate of two-day-old females parasitizing the eggs of the *D. angustalis* moth (55.04 ± 5.13%) is higher than that of one-day-old females, although the difference is not statistically significant (Figure 6F).

## 4. Discussion

Polyembryonic parasitoid wasps undergo complete metamorphosis, progressing through the following stages: eggs, larvae, pupae, and adults. They can complete one or more generations per year, with adult emergence occurring in spring, summer, or early autumn. For instance, *Litomastix heliothis* (Liao), a polyembryonic parasitoid of *Helicoverpa armigera* (Hübner), has one generation per year, with adults emerging in mid-April and peaking in late April, with the pupal stage lasting about 10 days. The larval stage is prolonged, lasting approximately 200 days, from early June to mid-March of the following year. Other polyembryonic species, such as an unidentified member of the same genus *Litomastix*, may complete up to 5–6 generations per year, with one generation taking about 28 days [38]. The overwintering form of these parasitoids is typically the larval stage, at which point hibernation occurs within the host larvae. Upon adult emergence, mating and egg-laying follow, with parasitoid larvae hatching and feeding on the host eggs. The egg–larval period lasts about 21 days, and the pupal stage is approximately 6 days. Different polyembryonic parasitoid species may have varying developmental durations and life cycles but generally synchronize closely with the life cycle of their hosts [38]. This study clarifies the morphological characteristics, developmental stages, and life history of *C. nacoleiae*. The average number of eggs laid by *C. nacoleiae* females in each *D. angustalis* egg was 8.47, and the number of eggs laid per *C. nacoleiae* life was 107.55 ± 28.38. These eggs underwent division and development through the larval and pupal stages, ultimately producing an average of 1489.47 adults. This result confirms that *C. nacoleiae* employs a polyembryonic reproductive strategy, making it a cross-stage parasitoid of the *D. angustalis* egg and larval stage. This reproductive strategy is similar to that of the *Copidosoma floridanum*, which parasitizes *Trichoplusia ni* [19,39].

Under laboratory conditions, over 99.59% of *C. nacoleiae* adults eclosed between 6:00 and 12:00, with only 0.82% of males eclosing at 19:00, and no females eclosed in the afternoon. The adults of *Coccophagus ceroplastae* (Howard), *Coccophagus yoshidae* (Nakayama) (Aphelinidae), and *Chrysonotomyia formosa* (Westwood) (Eulophidae) primarily eclosed between 8:00 and 10:00, which is similar to the eclosion rhythm observed in *C. nacoleiae* [40,41,42]. Adult eclosion occurs during the daytime in most parasitoid species, with a peak in the morning at suitable environmental temperatures. Both female and male *C. nacoleiae* have a peak eclosion time at 7:00, which differs from *C. formosa*, where males eclose earlier than females, with a more concentrated and distinct peak [41,43]. However, both sexes of *C. nacoleiae* have the same eclosion peak, making it easier for parasitoids to encounter each other for pairing in a timely manner, ensuring the optimal number of males and females for maximum reproductive efficiency.

Ode et al. [2] summarized that within Hymenoptera, polyembryony is found only in a few parasitic families, including Aphelinidae, Encyrtidae, Dryinidae, Braconidae, and Platygastridae. This is relatively uncommon among hymenopteran insects. However, the reasons and implications of polyembryony remain unclear. From an adaptive standpoint, polyembryony overcomes the limitation imposed by the number of eggs a female can lay, allowing a single or a few egg-laying events to produce a large number of offspring. Moreover, parasitoids that exhibit polyembryony are classified as koinobiont parasitoids, meaning that they do not immediately kill the host but instead suppress or delay its development, eventually leading to the host’s death. As a result, the number of offspring produced through polyembryony is regulated and aligns with the host’s capacity to support them [44,45,46,47]. In other words, *C. nacoleiae* is not constrained by the egg-laying capacity of the female parent but rather maximizes its population size based on the host’s carrying capacity. The number of *C. nacoleiae* offspring increases with the body length of the host’s mature larva. Segoli et al. suggested that the offspring of polyembryonic parasitoids might influence the size of the hatching population by regulating the division of embryos [48]. The “hatch-benefit hypothesis” supports a similar view: the number of divisions and hatchlings in polyembryonic parasitoids is controlled by the overall population size of the host, thus reducing the intensity of intraspecific competition [2]. Furthermore, as *T. evanescens* larvae cease feeding once they enter the mature stage, to foster larger wasp populations, sufficient food must be provided for the host larvae during rearing. This ensures ample nutrients in the mature larvae, which is crucial for producing a high number of *C. nacoleiae* offspring.

The lifespan of polyembryonic parasitoid wasp adults ranges from 1 to 20 days, with supplementary nutrition influencing their longevity. For example, the lifespan of *Litomastix heliothis* adults is 7–20 days [49] and that of *Litomastix truncatellus* adults is 8–10 days, extending up to 14 days with supplemental nutrition [50]. Zhang et al. studied the effect of different nutritional conditions on the lifespan of *Aenasius arizonensis* (Girault) (Encyrtidae), finding that the longest lifespan of up to 9 days occurred when the wasps were fed a 10% honey solution and mealybug honeydew. In contrast, the lifespan of adults of an unidentified *Litomastix* species is much shorter, only 1 day [51,52]. Thus, supplemental nutrition can extend the lifespan of polyembryonic parasitoid wasps. The lifespan of *C. nacoleiae* adults feeding on 10% honey water was found to be 2.18 days, which is consistent with the results of Zhang et al. [53]. Zhang et al. tested various nutritional conditions for *A*. *arizonensis*, and their findings showed that the longest lifespan was observed in individuals fed a combination of 10% honey water and mealybug honeydew [53]. Similarly, Zheng et al. tested nutritional conditions for *Anagrus nilaparvatae* Pang et Wang (Mymaridae), and their results also indicated that 10% honey water yielded the longest lifespan [54]. However, the optimal nutritional condition for *Leptopilina japonica* Novković et Kimura (Figitidae) was found to be 20% honey water, not 10% [54]. Different parasitoids have different optimal nutritional conditions, and using the appropriate concentration of nutrient solutions is the most effective approach to extend the parasitoid’s lifespan [55,56]. Therefore, 10% honey water is the best nutritional condition for maintaining the lifespan of *C. nacoleiae* adults. Additionally, it was found that females had higher early survival rates than males in the blank and distilled water treatments when comparing the survival rates of *C. nacoleiae* males and females under different nutritional treatments. In contrast, in 5–20% honey water treatments, males had higher early survival rates than females. This may be related to the shorter lifespan of females under nutritional treatment conditions.

## 5. Conclusions

In the laboratory, the average offspring production of *C. nacoleiae* larvae was 1489.47 ± 77.54 individuals, with a maximum of 2024 individuals. This production increased with the body length of the host’s mature larvae. Therefore, when artificially rearing host larvae, it is crucial to provide sufficient food to ensure their healthy development and maximum body length, which, in turn, will increase the number of emerging adults. Regarding the adults under nutritional treatment conditions, those fed a 10% honey water solution had the longest lifespan, averaging 2.18 ± 0.09 days. Hence, when rearing *C. nacoleiae* in the laboratory, it is recommended that a 10% honey water solution be provided as the primary nutritional source for the adults. In terms of oviposition behavior, *C. nacoleiae* females mainly lay eggs between 10:00 and 15:00. The oviposition rate of 1-day-old females peaks around 12:00, reaching 88.89 ± 5.37%, significantly higher than that of 2-day-old females. Furthermore, when the ratio of females to host eggs is 1:10, the parasitism rate of *C. nacoleiae* is the highest, averaging 92.72 ± 2.92%. Additionally, mating behavior in *C. nacoleiae* adults occurs primarily between 10:00 and 13:00. The mating rate of 2-day-old adults peaks around 10:00, reaching 38.89 ± 5.37%, significantly higher than that of 1-day-old adults. Oviposition by 2-day-old females is also concentrated between 10:00 and 15:00. Based on the reproductive rhythm of *C. nacoleiae*, it is recommended that females are released early in the morning (before 10:00 a.m.) during the egg-laying period of *C. nacoleiae* when using this species for biological control in the field. This ensures the highest parasitism rate and the best control efficacy. Conversely, when using chemical pesticides, it is advised to avoid applying them between 10:00 and 16:00 to minimize the impact on the natural populations of *C. nacoleiae*.

## Figures and Tables

**Figure 1 insects-16-00239-f001:**
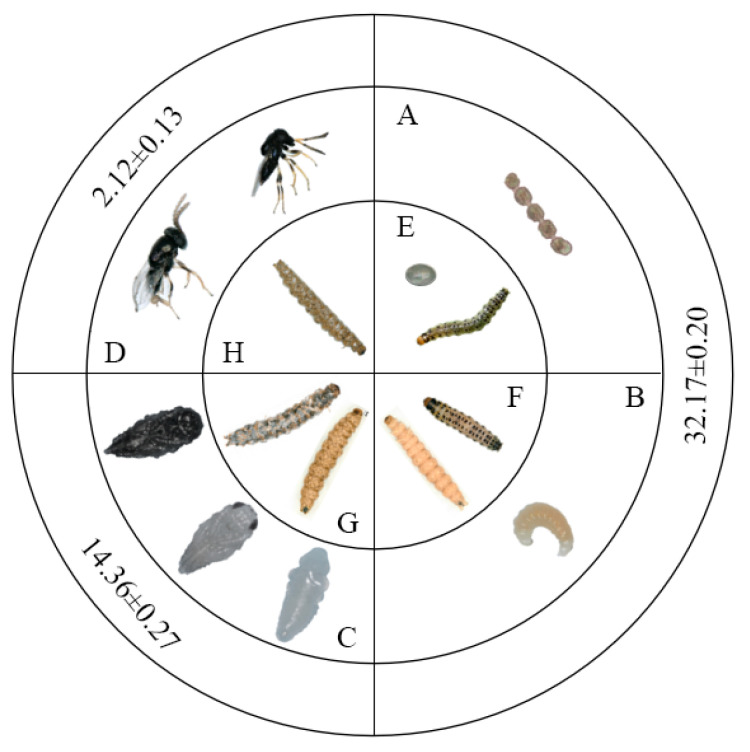
Life cycle of *C. nacoleiae*. (**A**): eggs; (**B**): larva; (**C**): pupae (early, middle, and late stages, shown clockwise, same as below); (**D**): adult; (**E**): an egg and a larva of the host; (**F**): intact and parasitized larvae of the host; (**G**): hosts at early and late stages of *C. nacoleiae* pupae; (**H**): host shell after emergence of *C. nacoleiae*. Data (mean ± SE) represent the duration of the stages of *C. nacoleiae*, measured in days.

**Figure 2 insects-16-00239-f002:**
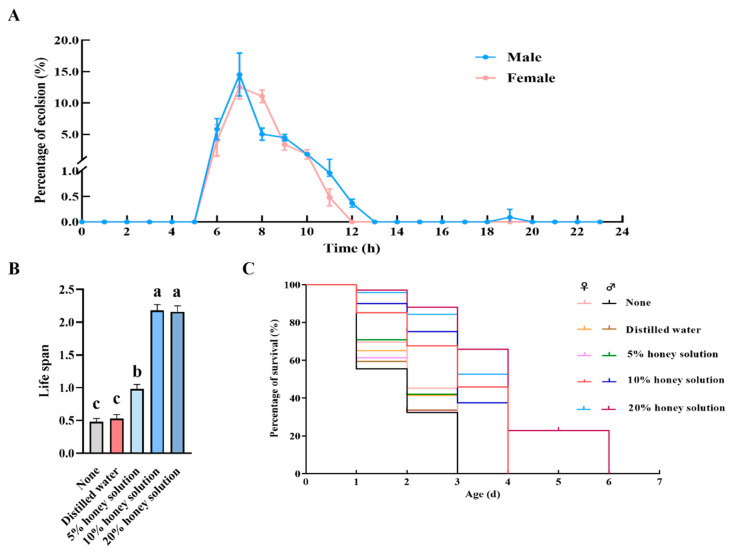
Life habits of *C. nacoleiae* adults. (**A**) Circadian rhythm of newly emerged adult *C. nacoleiae*. (**B**) Effects of nutritional supplementation on lifespan of adult *C. nacoleiae* (measured in days). (**C**) Survival curves of adult *C. nacoleiae* with different nutritional treatments. Different lowercase letters indicate significant differences between different nutritional supplementation (*p* < 0.05).

**Figure 3 insects-16-00239-f003:**
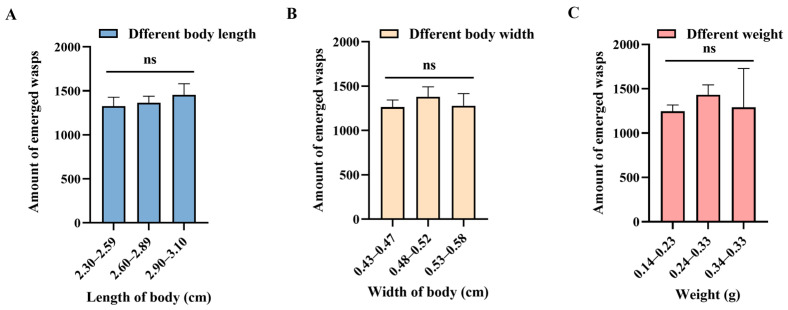
The number of emerging *C. nacoleiae* is related to the host’s body length, body width, and body weight. (**A**) Number of emerged wasps from mature larval hosts of different body lengths. (**B**) Number of emerged wasps from mature larval hosts of different body widths. (**C**) Number of emerged wasps from mature larval hosts of different weights. ns signifies no significance.

**Figure 4 insects-16-00239-f004:**
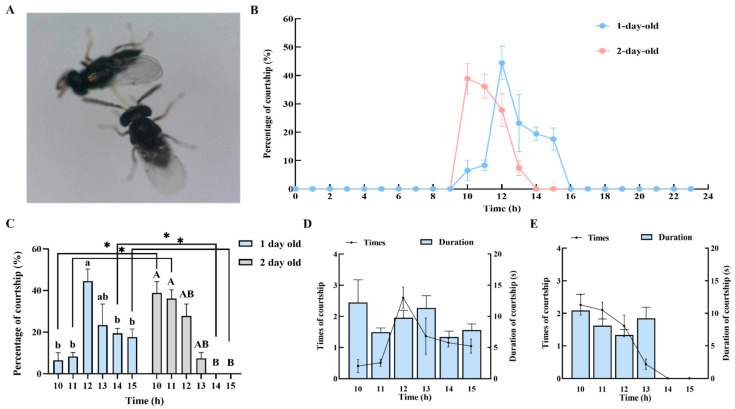
Courtship behavior and rhythms of male *C. nacoleiae*. (**A**) Male *C. nacoleiae* searching for female hosts. (**B**) Courtship rhythm of male *C. nacoleiae* of different ages. (**C**) Differential analysis of courtship percentages of male *C. nacoleiae* of different ages. * indicates that there are significant differences in males of different ages (*p* < 0.05). (**D**) Times and duration of courtship of 1-day-old male *C. nacoleiae*. (**E**) Times and duration of courtship of 2-day-old male *C. nacoleiae*. Different lowercase letters indicate significant differences between 1-day-old adults at different times. Different capital letters indicate significant differences between 2-day-old adults at different times (*p* < 0.05).

**Figure 5 insects-16-00239-f005:**
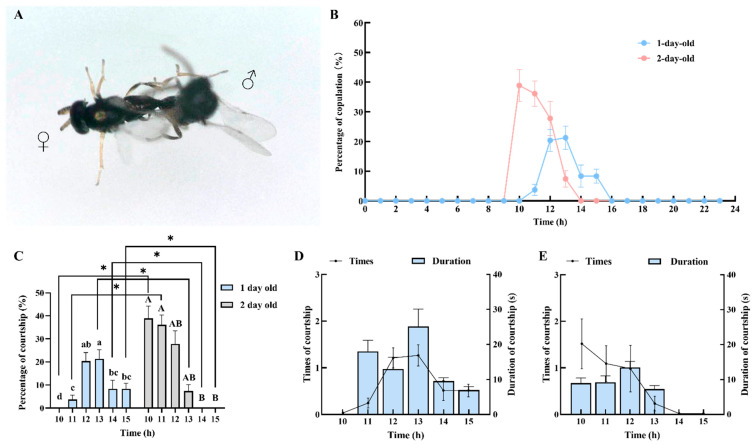
Mating behavior and rhythms of *C. nacoleiae* adults. (**A**) Copulation of adult *C. nacoleiae*. (**B**) Copulation rhythm of *C. nacoleiae* of different ages. (**C**) Results of differential analysis of copulation percentages of adult *C. nacoleiae* of different ages. * indicates that there are significant differences in adults of different ages (*p* < 0.05). (**D**) Times and duration of copulation of 1-day-old *C. nacoleiae*. (**E**) Times and duration of copulation of 2-day-old *C. nacoleiae*. Different lowercase letters indicate significant differences between 1-day-old adults at different times. Different capital letters indicate significant differences between 2-day-old adults at different times (*p* < 0.05).

**Figure 6 insects-16-00239-f006:**
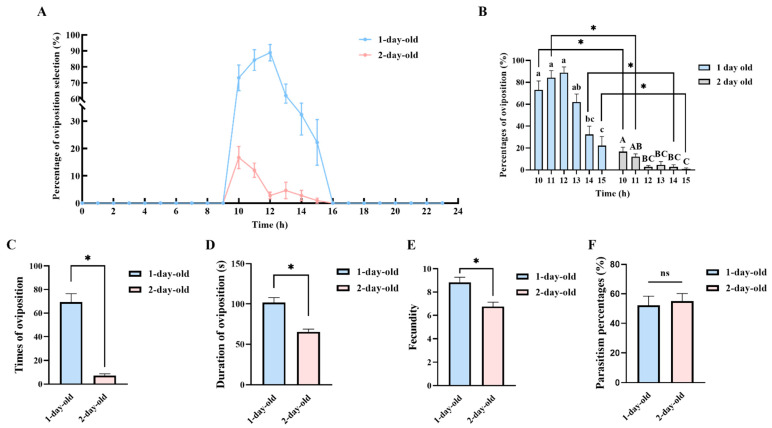
Oviposition behavior and rhythms of female *C. nacoleiae*. (**A**) Oviposition rhythm of *C. nacoleiae* of different ages. (**B**) Differential analysis of *C. nacoleiae* female oviposition rate at different ages. Different lowercase letters indicate significant differences between 1-day-old adults at different times. Different capital letters indicate significant differences between 2-day-old adults at different times (*p* < 0.05). (**C**) Times of oviposition of female *C. nacoleiae* of different ages. (**D**) Duration of oviposition of female *C. nacoleiae* of different ages. (**E**) Fecundity of female *C. nacoleiae* of different ages. (**F**) Parasitism percentages of female *C. nacoleiae* of different ages. * indicates that there are significant differences in females of different ages (*p* < 0.05). ns indicates that there is no significant difference in females of different ages.

## Data Availability

The data presented in this study are available on request from the corresponding author.

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
