# Peer review of "Reproductive Behavior of the Polyembryonic Parasitoid Copidosomopsis nacoleiae (Eady) at Different Ages"

_insects, 2025, doi:10.3390/insects16030239_

Round 1

Reviewer 1 Report

Comments and Suggestions for Authors

I just reviewed the manuscript  Insects-341994 titled “Reproductive behavior of the polyembryonic parasitoid Copidosomopsis nacoleiae at different ages”. 

The objective of the work is very interesting, also because the biology of the polyembryonic wasps is a complicate and peculiar topic. 

However, the manuscript has several critical issues and it is not acceptable.  I am not sure if the problem was due to the poor English knowledge (the manuscript needs an important revision of the language), but the organization of the experiments is very confusing. I noticed that there are a lot of missing data, starting from the beginning of the Introduction. 

This is the list of the critical aspects:

-) The introduction is too long and very dispersive; I understand that it is correct to introduce the polyembryonic wasps in a holistic way, but at the end the Authors wrote almost 2 pages on polymbryonic wasps and very little on the target species, Copidosomopsis nacoleiae. 

-) Moreover, when Authors start to introduce the target species, they forgot to follow correct Entomological nomenclature code: the first time you are reporting a species, you should write the full genus name, the species name and the Author(s) name(s): well, the full genus name and the Author’s name (of the insect subject of the Research) are missing!

-) Still in the introduction chapter, very little is written on the biology of Cnacoleiae. In particular -since the title of the manuscript is focused on the “reproductive behavior” the Authors forgot to report the oviposition behavior of the species (the wasp female is laying eggs in the egg of the host moth). Since some of bioassays were addressed to study the fertility and fecundity of the wasp species, the description of this part is extremely important.

-) Last point in the introduction: Authors wrote a part describing the field work done from 2018 to 2020 in a specific area of China, describing the occurrence of the host species (the moth D. angustalis) defoliator on an important plant (A. scholaris). Very interesting, but the reported data are not enough: Authors are telling that “only 25.33 larvae (were) observed in December”. What does it means? And on what? A tree, a leaf, a brunch? If 25.33 is lower limiting factor, what would be the correct parameters in high infestation conditions? Are these studies published, or are they unpublished data (in any case a table showing the recorded data, would be useful)? 

-) The situation in the Material and Methods chapter is not very different: missing data on when the experiments were carried out; poor descriptions of the rearing of the larvae of the parasitized moths; in the Sub-Chapter 2.2.1. Authors described a bioassay using a sample of the eggs to perform morphological studies on the eggs: I did not find any output on the results of this part. Most of the bioassay descriptions are not very clear: for sure there are problems with the language, but also it seems that the Authors misunderstood the importance to explain why a specific research is done, providing all the details of the experimental design. Sometimes the experimental conditions were not adequate (or at least not well described).

-) Moving to the Tesults, Authors did not report the sex ratio of the emerging wasp adults, important factor to evaluate the quality of the rearing. Authors mentioned observations on the courtship behavior: actually, they just recorded the number of events. Also in other bioassays, the work done is poorly described. 

Comments on the Quality of English Language

The English and the organization of the paragraphs must be reviewed by English native mother tongue: several time is difficult to understand what the Authors would like to explain.

Author Response

Dear editors and reviewers,

We tried our best to improve the manuscript entitled “Reproductive Behavior of the Polyembryonic Parasitoid Copidosomopsis nacoleiae (Eady) at different ages” (ID: insects-3419964) and made some changes to the manuscript. These changes will not influence the content and framework of the paper. As you are concerned, several issues need to be addressed, which are replied to in detail as below.

Response to the comments for Reviewer #1

Dear reviewer #1,

According to the constructive advice, we have made extensive modifications to our manuscript and supplemented extra data to make our results convincing. Thank you again for your positive comments and valuable suggestions to improve the quality of our manuscript.

The detailed corrections are listed below.

Q1. Title: The name of "Copidosomopsis nacoleiae" is not complete.

Response: Thank you for your reminder. The name of "Copidosomopsis nacoleiae" has been changed to "Copidosomopsis nacoleiae (Eady)".

Q2. Simple Summary: "C. nacoleiae" and "D. angustalis" appear for the first time without giving their full names.

Response: We were really sorry for our careless mistakes. "C. nacoleiae" was replaced by "Copidosomopsis Nacoleiaenacoleiae (Eady) (Hymenoptera: Encyrtidae)", and "D. angustalis" was replaced by "Diaphania angustalis (Snellen) (Lepidoptera; Crambidae)" (line 13-17).

Q3. Lines 60-61. Introduction: there is a discrepancy between the authors mentioned at the beginning (Hu et al.) and the reference numbe ([5]).

Response: We apologize for our errors in citing literature, the references have been corrected (line 64).

Q4. Line 71-72. This sentence is written wrong: at the beginning authors wrote "a single host can produce hundreds...parasitoids" and then as exapìample they mentioned C. floridanum (a polyembryonic parasitoid) can produce up to 1,100 parasitoids... Please modify the text.

Response: It is expressed as "A single host can produce hundreds or even thousands of parasitoids", which is not contradictory to "C.floridanum can produce up to 1,100 parasitoids".

Q5. Line 82. Scientifically wrong, change in "different aphid species".

Response: Done.

Q6. Line 89-90. According to what I can understand, this sentence is not connected with the previous ones: if I am correct Authors should move down, starting a new paragraph. Moreover, it is not well written: "... and the genus Diaphania Hubner" is not connected with the text.

Response: We sincerely appreciate the valuable comments. We have moved this sentence down and started a new paragraph. The paragraph begins with "The parasitoid Copidosomopsis nacoleiae (Eady) is a polyembryonic endoparasitoid wasp recently discovered to parasitize the egg-larvae of D. angustalis" (line 89).

Q7. Line 86. "Alstonia scholaris (L.), commonly" lacks family introduction

Response: Done.

Q8. Line 93. "C. nacoleiae" lacks full genus name, Author and other taxonomic records (IT IS THE FIRST TIME IN THE MANUSCRIPT WHEN THE THE INSECT THAT THEY STUDIED IS MENTIONED)

Response: Thank you for the highly valuable comment. "C. nacoleiae" was replaced by "Copidosomopsis Nacoleiaenacoleiae (Eady)".

Q9. Line 106. "... with only 25.33 ± 22.40 larvae observed in December." there is not any indication about the sampling methods: so, writing that you recorded 25.33 larvae without any information on the methods (on how many leaves? how many larvae were present in high infestation periods?) is not suitable: please change or delete. Moreover, it is not clear if these data are a part of the work that the Authors would like to present of if they are just general background information: in the first case, Authors must move all to the next chapters; if it is true the second hypothesis, they must include reference notes for the bibliographic records..

Response: The field population dynamics of D. angustalis from 2018 to 2020 are obtained by our investigation. Our purpose is to use this part of the data as background content to give readers a better understanding of the occurrence and damage of D. angustalis and the occurrence of C. nacoleiae in the field.

Q10. Line 119-120. What is the meaning of the word "systematic" in this context? written wrongly: it should be "wasp males"

Response: I'm sorry for my unprofessional expression, and I'm grateful that you can understand my expression. The original sentence has been replaced by "The courtship behavior of parasitic wasps follows the usual pattern, wasp males court females by repeatedly tapping the females with their antennae in a series of patterned actions".

Q11. Line 151. It is not clear to me the meaning of this word in the context... completion of what?

Response: Thank you very much for your professional reminder, we are sorry for the confusing description of the oviposition behavior. "completion" means the end of the spawning process. In order to avoid misunderstanding and concise description, we delete "completion".

Q12. Line 179-186. Materials and Methods: When the material has been collected in the field (months/years)? If possible provide some additional information on the numbers (for each host stage). Some evident mistakes in English..Additional effort must be given to describe the rearing of the moth... in particular, it will be important to describe the oviposition site (leaves? paper? later Authors are mentioning that 300 eggs of the moth have been used to study some aspects of the oviposition of the parasitoid: does the moth female oviposit a cluster of eggs? if this is the case, did the authors choose eggs from a single female cluster?

Response: We strongly endorse your professional insights. Materials were collected in the field every month from 2018 to 2020. D. angustalis oviposits on plastic cups or leaves. Meanwhile, the moth female oviposit a cluster of eggs. However, the egg clusters used in the experiment were not from a single female.

Q13. Line 186. "D. angustalis" italics

Response: Done.

Q14. Line 192-196. How many parasitized pupae? From where the parasitized material is coming from? Field or laboratory? if field, specify the collection site and the period of the year when they have been collected. In the last sentence, it seems that there is a technique to sex the parasitoid wasp at the pupal stage: if it is a normal technique using morphological parameters, authors should describe better the protocol, otherwise they mast report the bibliographic reference referred to to the technique.

Response: Thank you very much for your insight, the number of parasitized pupae is mentioned in the method of each experiment later. The parasitized materials are all the eggs produced by the reproduction of D. angustalis in the laboratory. The original text "parasitized mature larvae of D. angustalis were identified and individually reared" has been replaced by "60 parasitized mature larvae of D. angustalis cultured in the laboratory were identified and individually reared." I am sorry for the incorrect gender identification description in the original text. We distinguish male and female according to the number of antennae of mature pupae, which is described in more detail in another article that is being submitted. In short, the female antennae are 9 segments in length, while the male antennae are 10 segments in length (line 205-214).

Q15. Line 200. Fed with a 10%....

Response: Done.

Q16. Line 204-205. See my comments in the previous page; Please describe the size and the type of cage. 

Response: Thank you for your remainder. The cage size is 10*10*30 cm, and the description of the size of the cage is also added in the original text (line 224).

Q17. If I understood properly, D. angustalis is the target pest: correct? Second questoion: the 30 males and 30 females wasps are the parasitoid C. nacoleine, correct? So, if this is correct, it seems that the parasitoid C. nacoleine is an egg parasitoid (or at least oviposit into the egg of his host: correct? This type of information must be described muche better into the introduction: it is not acceptable that the readers will try to understand the life cycle without any details from the Authors.

Response: We sincerely thank you for your guidance. Your comments are of great help to improve the quality of our papers. D. angustalis is the target pest, and the parasitoid C.nacoleiae is the egg parasitoid of the pest. In the introduction, the relevant description is added to facilitate readers to better understand the life cycle of the C.nacoleiae (line 225-228).

Q18. Line 224. Change "Rhythm" to "Circadian rhythm".

Response: Done.

Q19. Line 228-229. Why? what is the reason Authors are doing a monitoring of the host, after removing the parasitoid pupae from it?

Response: Observing the host is because it is necessary to predict when the parasitoid may emerge, start recording and observing the adult emergence behavior of the parasitoid a little earlier.

Q20. Line 230-234. It is very difficult to understand what is written and the protocol.  According to what I guess, 25 lep parvae were added afterwards... why?

Response: Instead of adding 25 lep parvae, 25 hosts that were parasitized and had identified the sex of the parasitic wasps were cultured separately to observe the adult emergence behavior.

Q21. Line 236. Impossible to accept withou providing more details: size, shape, material, ventialtion system). According to the decription is a vary forced context and insects are showing unusual wrong behaviors in such strong captivity conditions.

Response: I' m sorry that my carelessness led to a bad reading experience. The place where the male and female are paired is not pipette, but 5 mL finger tube (line 256).

Q22. Line 266. Dimension of the Petri Dish. 

Response: Done. The diameter of Petri Dish is 9 cm.

Q23. Line 278-284. Poor description of the methods: are females virgin? I am assuming that the experiment is connected with the previous one, but must be mentioned. I was also not reported the type of confining sytem used and the climatic conditions. 

Response: Females are virgin. All experiments were carried out under the conditions of temperature 25 ± 2 ℃, relative humidity 70 ± 5 %, photoperiod 14L: 10D. The pairing, repetition and observation methods were the same, that is, one female and one male of C.nacoleiae at the same age (1 day old female and 1 day old male, 2 day old female and 2 day old male) were paired in a petri dish, and 10 eggs of D.angustalis and 10 % molasses water were given, repeated 30 times. The behavior was continuously observed for 24 hours with a super-depth-of-field three-dimensional microscopic system, and the video was recorded with a high-definition camera.

Q24. Line 287. Please provide information how this bioassay has been carried out.

Response: I' m sorry that my carelessness led to a bad reading experience. I very much agree with the reviewer 's point of view and revised the original text.

Q25. Line 287. Change "when a mated female resting on" to "when the female paused".

Response: Done.

Q26. Line 378-379. Not molting is observed: no host larval molting or not wasp larval molting? if the second is correct, it would be wrong to write "remain active and develop" (there was not larval development until that stage.

Response: We have re-written this part according to the reviewer’s suggestion. We deleted " During this period, no molting is observed, so the larvae are not staged by age". The larval stage of C.nacoleiae was active and developed in the host. We did not instar the larvae during the experiment (line 405). 

Q27. Line 379-385. it is really not clear! It is an important part and must be developed with more details.

Response: We have added more details.

Q28. Line 391. change "rhythms" in "circadian rhythm".

Response: Done.

Q29. Line 530. This sentence is not very clear: my suggestion is to provide addititional information. Female wasps are laying only 8.47 eggs in their life spam?

Response: The average number of eggs laid by C. nacoleiae females in each D. angustalis egg was 8.47, and the number of eggs laid per C. nacoleiae life was 107.55 ± 28.38 (line 561-562).

Q30. Line 573. Adults or full life cycle?

Response: Adults. We replace the original sentence with "The lifespan of polyembryonic parasitoid wasp adults ranges from 1 to 20 days" (line 604).

We would like to express our appreciation to you and reviews on our manuscript and are looking forward to your reply. We hope that the revision can meet the requirements of the journal for review.

Best regards,

Huili Ouyang, Xiaoyun Wang

Jan 26th, 2025

Reviewer 2 Report

Comments and Suggestions for Authors

Dear Dr. Wang,

I have carefully read your manuscript entitled "Reproductive Behavior of the Polyembryonic Parasitoid Copidosomopsis nacoleiae at different ages". I believe it contains new information about the reproductive behavior of different developmental stages of C. nacoleiae, and therefore the manuscript could be published in Insects. However, I suggest a few corrections to the text. First, Introduction seems overloaded with detailed references to papers of various authors, which, I believe, are much more appropriate in the Discussion section. Second, the authors correctly treat C. nacoleiae as a member of the genus Copidosoma throughout the whole paper. Nevertheless, it is referred to as Copidosomopsis in the title and keywords. According to the latest version of the Universal Chalcidoidea Database (https://ucd.chalcid.org), Copidosomopsis is currently considered a synonym of Copidosoma. Other suggestions can be found in the attached .pdf file.

Comments on the Quality of English Language

The language of the manuscript must be improved. Although I do have some suggestions in terms of language quality (please see the attached .pdf file), I also recommend an extensive revision of the text, preferably by a native English speaker.

Author Response

Dear editors and reviewers,

We are very grateful for your constructive comments and suggestions for our manuscript entitled “Reproductive Behavior of the Polyembryonic Parasitoid Copidosomopsis nacoleiae (Eady) at different ages” (ID: insects-3419964). Your comments are very valuable and helpful for improving our manuscript. We have tried our best to make all the revisions clear, and we hope that the revised manuscript can satisfy the requirements for publication. The main revisions in the new manuscript are as flowing:

Response to the comments for Reviewer #2

Dear reviewer #2,

We would like to thank you for your professional review work, constructive comments, and valuable suggestions on our manuscript. According to your nice suggestions, we have made extensive corrections to our previous draft, the detailed corrections are listed below.

Q1. Simple Summary: Change "polyembryonic" to "multi-embryo"

Response: Thank you for your insightful comment. Done.

Q2. Simple Summary: Please give the genus name in full, and add the names of the order and family.

Response: We sincerely thank the reviewer for careful reading. As suggested by the reviewer, we have revised the simple summary based on the reviewer's comments.

Q3. Line 23: Please add the names of the order and family.

Response: Thank you for your remainder. We have added the names of the order and family.

Q4. Line 45: If you refer to a particular expert, you must cite his work

Response: Done.

Q5. Lines 45; Replace "proposed" with "discovered".

Response: Done.

Q6. Lines 53: Do you mean "cannibalism between the embryos"?

Response: Thank you for understanding our inappropriate description. We have changed "intra-embryo cannibalism" to "cannibalism between the embryos".

Q7. Lines 61: I believe it would be useful if you add family names at the first mention of each parasitoid.

Response: We think this is an excellent suggestion. We have added family names at the first mention of each parasitoid.

Q8. Lines 67: "Dalm" misspelled, changed to "Dalman".

Response: Done.

Q9. Lines 82: Change "the aphid Aphididae sp." to "aphids".,

Response: Done.

Q10. Line 85: I recommend to start it from a new paragraph

Response: Thanks to the reviewer 's suggestion, we changed "Diaphania angustalis (Snellen) " to "D. angustalis" and start it from a new paragraph.

Q11. Line 107: add "of".

Response: Done.

Q12. Line 118: Do you mean something like "follows the usual pattern"?

Response: Yes. In order to make readers better understand, we use the reviewer 's description. We change "is systematic" to "follow the usual pattern".

Q13. Line 122: Delete " ,using its antennae to control hers from the outside"

Response: Done.

Q14. Line 146: Delete "post-mating"

Response: Done.

Q15. Line 157: [32] It seems the numbers of this and some other references in this paragraph are shifted. Please check whether these numbers correspond to those in the reference list.

Response: Thanks to the reviewer, we have checked the cited literature.

Q16. Line 162-163: Change "Telenomus (Aholcus) lebedae" to "Telenomus lebedae (Aholcus)"

Response: Done.

Q17. Line 165: Judging from its preferred host, Lasioderma serricorne, these parasitoids may actually belong to Anisopteromalus quinarius Gokhman et Baur (see Baur et al. 2014, https://doi.org/10.1111/syen.12081)

Response: References have been replaced.

Q18. Line 181: Change "petri dishes" to "Petri dishes".

Response: Done.

Q19. Line 182, 183, 200: Lack of preposition "with" after "feed".

Response: We carefully examined the full text and corrected similar errors one by one.

Q20. Line 187: Please give "D. angustalis" in italics.

Response: Done.

Q21. Figure 1: Change "hosts" to "the host".

Response: Done.

Q22. Figure 1: Change "a mature larva of hosts and the parasitized appearance" to "an intact and parasitized larvae of the host" .

Response: Done.

Q23. Figure 1: Change "the unit of data was day" to "measured in days".

Response: Done.

Q24. Line 394: Change "Fig. 4A" to "(Fig. 4A)".

Response: Done.

Q25. Figure 4: Change "different age male C. nacoleiae" to "male C. nacoleiae of different age"

Response: Done.

Q26. Line 522: Change "Litomastix sp" to "an unidentified member of the same genus Litomastix"

Response: Done.

Q27. Line 546: Change "to collect parasitoids" to "for parasitoids to encounter each other"

Response: Done.

Q28. Line 549: What about the Braconidae?

Response: We have added information about Braconidae.

Q29. Line 567-568: The main features of this evolutionary strategy are essentially the same for all polyembryonic parasitoids, and they are not restricted just to C. nacoleiae.

Response: We sincerely thank the reviewer for careful reading. We had changed "which suggests that C. nacoleiae may exhibit an unusual evolutionary" to "which suggests that C. nacoleiae may also exhibit this evolutionary strategy". (line 607-608)

Q30. Line 580: Change "Litomastix sp." to "an unidentified Litomastix species".

Response: Done.

Q31. Line 584: Delate "and".

Response: Done.

Q32. Line 586: "Zheng et al. (2003) tested nutritional conditions for Anagrus nilaparvatae..." I do not see this reference in the list.

Response: .

Q33. Line 588: Please give "Leptopilina japonica" in italics.

Response: Done.

Q34. Line 611: Add "the".

Response: Done.

Q35. Line 616-617: Change "females when using C. nacoleiae" to "when using this species".

Response: Done.

Q36. References 9: Change "Eneyrtidae" to "Encyrtidae".

Response: Done.

Q37. References 14: Change "." to ",".

Response: Done.

Q38. References 17: Please give " Copidosomopsis" in italics.

Response: Done.

Q39. References 36: Change "Litomactix" to "Litomastix".

Response: Done.

Q40. References 48: Change "Polyembryonic wasp" to "the polyembryonic wasp".

Response: Done.

Q41. References 50: Change "HymenoPtera: Chaleidoidea" to "Hymenoptera: Chalcidoidea".

Response: Done.

We appreciate for reviewers’ warm work earnestly, and hope the correction will meet with approval. Once again, thank you very much for your comments and suggestions.

Best regards,

Huili Ouyang, Xiaoyun Wang

Jan 26th, 2025

Round 2

Reviewer 1 Report

Comments and Suggestions for Authors

I made the second revision of the manuscript insects-3419964. There are rules that must be followed for the Simple Summary, the Abstract and the Introduction: in all of them , when a scientific name of an insect or a plant is mentioned for the first time, the genus and the species names should be fully written. In the introduction (as well in the Discussion or Conclusions chapters, in addition to the full name, the first time an insect or a plant is reported also the Author's name and order and family should be included. 

If a sentence is starting with the scientific name, it is better to re-write the full genus name again.

A part of the introduction is very confusing. Please see my comments in the attached file.  Authors are mentioning that the parasitoid is also developing in aphids, reporting several scientific names of insects that are NOT aphids!

Several sentences are written in the wrong context:  the feeling is that a "copy and past" approach without following a logic order.  

The introduction is too long, especially when is provide a pot of general information on polyembryonic parasitoids: still  alo of more important parts on the life cycle of the host are missing. The last part of the introduction is (instead) too short: this part is essential to describe better the purpose of the work. Definitely must be improved!

Also this observation is valid for Material and Methods chapter. 

Comments on the Quality of English Language

I am not an English native person, but the feeling is that the English should be a little bit more  improved (but it is better than in the first version).. 

Author Response

Dear editors and reviewers,

We are very grateful for your constructive comments and suggestions for our manuscript entitled “Reproductive Behavior of the Polyembryonic Parasitoid Copidosomopsis nacoleiae (Eady) at different ages” (ID: insects-3419964). Your comments are very valuable and helpful for improving our manuscript. We have tried our best to make all the revisions clear, and we hope that the revised manuscript can satisfy the requirements for publication. The main revisions in the new manuscript are as flowing:

Response to the comments for Reviewer #1

Dear reviewer #1,

We would like to thank you for your professional review work, constructive comments, and valuable suggestions on our manuscript. According to your nice suggestions, we have made extensive corrections to our previous draft, the detailed corrections are listed below.

Q1. Simple Summary: in the "Simple Summary" (as well in the "Abstract") there is not need to report the Author's name of the insect as well the order/family (BUT IT IS VERY IMPORTANT TO REPORT THEM IN THE REAL MANUSCRIPT BELOW).

Response: Thank you for your insightful comment. Done.

Q2. Abstract: The "Simple Summaty" and the "Abstract" should be considered separate entities among them and with the full manuscript below: so, the full genus name of the parasitoid and its host must be reported also in the Abstract.

Response: We sincerely thank the reviewer for careful reading. As suggested by the reviewer, we have revised the full genus name of the parasitoid and its host based on the reviewer's comments.

Q3. Line 85. Introduction: this is not correct: all the following scientific names do not have anything to do with aphid species! My suggestion is to erase the sentence, from "and other aphid species, unti (guen) [10, 11]". 

Response: Thank you for your remainder. We have deleted this sentence as suggested by the reviewer.

Q4. Line 87-90: This long sentence should be reduced and moved down, after the next sentence that should be shorter in this way: "The parasitoid Copidosomopsis nacoleiae (Eady) (Hymenoptera: Encyrtidae) is a polyembryonic endoparsitoid wasp.  Its host range includes pests from Pyralidae, Crambidae, and Olethreutidae families within the Lepidopetra order. These pests  include Cnaphalocrocis. patnalis (Bradley) and C. ruralis (Walker), N. octasema (Meyr.), G. vertumnalis, L. octasema, and , recently, Diaphania angustalis (Snellen). Diaphania angustalis is one of the most significan defoliatord of Alstonia....

Response: We modified the sentence according to the reviewer 's suggestion.

Q5. Lines 102-105: as the previous comment (erase and add the references at the end the period reported in my comment above, referred to the host range of the moth.. ; erase this sentence and add the bibliographic reference above (see my previous cooment).

Response: We delete "Furthermore, C. nacoleiae is an egg-larva parasitoid wasp of ... folder Cnaphalocrocis medinalis (Guenée) and Nacoleia octasema (Meyr.) [19-21]." and merge it with the first sentence of the paragraph. We also corrected the corresponding references in the article.

Q6. Lines 178: larvae?

Response: Thank you for understanding our inappropriate description. We have added "larvae" before "D. angustalis".

Q7. Lines 187: What does it means? the tubes have been used to collect the parasitoids? it is not clear.

Response: 10 m L finger tubes were used to collect parasitic wasps.

Q8. Lines 196: sexed? in this case, I am suggesting to change in "sexually identified"..

Response: We have changed "identified" to "sexually identified".

Q9. Lines 196-197: Bibliographic record?

Response: Yes. 

We appreciate for reviewers’ warm work earnestly, and hope the correction will meet with approval. Once again, thank you very much for your comments and suggestions.

Best regards,

Huili Ouyang, Xiaoyun Wang

Feb 9th, 2025

Reviewer 2 Report

Comments and Suggestions for Authors

Dear Dr. Wang,

I have carefully read the revised version of your manuscript entitled "Reproductive Behavior of the Polyembryonic Parasitoid Copidosomopsis nacoleiae at different ages". As far as I can see, some substantial problems with the paper are solved now, and it could eventually be published in Insects. However, I still have a number of suggestions aimed to improve the manuscript. In addition to certain misprints, there are a couple of unclear statements in the text, which sometimes also does not meet the necessary taxonomic requirements (please see the attached PDF file). I believe the publication of the paper is possible only after the thorough check of the manuscript, which would provide the correct numbers in the text for the cited references, together with removing other errors.

Author Response

Dear editors and reviewers,

We tried our best to improve the manuscript entitled “Reproductive Behavior of the Polyembryonic Parasitoid Copidosomopsis nacoleiae (Eady) at different ages” (ID: insects-3419964) and made some changes to the manuscript. These changes will not influence the content and framework of the paper. As you are concerned, several issues need to be addressed, which are replied to in detail as below.

Response to the comments for Reviewer #2

Dear reviewer #2,

According to the constructive advice, we have made extensive modifications to our manuscript and supplemented extra data to make our results convincing. Thank you again for your positive comments and valuable suggestions to improve the quality of our manuscript.

The detailed corrections are listed below.

Q1. Abstract: Delete "this study systematically examines the" .

Response: Done.

Q2. Lines 58. Introduction: Delete "(1998)" .

Response: Done.

Q3. Lines 69. Introduction: According to modern classification, this species belongs to the genus Copidosoma. If you agree, please correct this throughout the whole text.

Response: Thank you for the highly valuable comment. We have correctted this throughout the whole text.

Q4. Line 87. Change "Fabr" to "Fabricius".

Response: Done .

Q5. Line 88-92. Scientifically wrong, change in "different aphid species".

Response: Done.

Q6. Line 90. Delete "the".

Response: Done.

Q7. Line 124. Add "(Braconidae)".

Response: Done.

Q8. Line 129. Add "(Braconidae)".

Response: Done.

Q9. Line 133. According to modern classification, the valid name of this species is Copidosoma primulus (Mercet). If you agree, please change the name throughout the text.

Response: Thank you very much for your professional reminder. We have changed the name throughout the whole text.

Q10. Line 134. Add " (Trichogrammatidae)".

Response: Done.

Q11. Line 151. Add " (Eulophidae)".

Response: Done.

Q12. Line 152. Add " (Cerambycidae)".

Response: Done.

Q13. Line 159. Add " Chen et Tong (Scelionidae)".

Response: Done.

Q14. Line 161. Please check once again throughout the whole manuscript whether the numbers in the text correspond to those in the reference list!

Response: We apologize for our errors in citing literature, the references have been corrected.

Q15. Line 161. (Pteromalidae; however, judjing from its biology, this must be A. quinarius Gokhman et Baur).

Response: Done.

Q16. Line 238. Change "wasp" to "wasps".

Response: Done.

Q17. Line 330. Change "they emerge as adults" to "adults emerge from them".

Response: Done.

Q18. Line 332. Change "on" to "in".

Response: Done.

Q19. Line 420. Change "comparing" to "compared".

Response: Done.

Q20. Line 445. Change "comparing" to "compared".

Response: Done.

Q21. Line 447. Delete "comparing courtship durations between different ages shows that".

Response: Done.

Q22. Line 510. Delete ",".

Response: Done.

Q23. Line 529. Change "offemale" to "of female".

Response: Done.

Q24. Line 541. Please give "Litomastix" in italics.

Response: Done.

Q25. Line 559. Add " (Aphelinidae)".

Response: Done.

Q26. Line 560. Add " (Eulophidae) ".

Response: Done. 

Q27. Line 562. Delete "adults".

Response: Done.

Q28. Line 569. change "(2018)" to "[2]".

Response: Done.

Q29. Line 570. change "Hymenopteran" to "hymenopteran".

Response: Done.

Q30. Line 571. Delete "for".

Response: Done.

Q31. Line 581. Delete "(2009)".

Response: Done.

Q32. Line 582. Again, please check these numbers throughout the whole text!.

Response: We apologize for our errors in citing literature, and we have exchanged references [47] and [48].

Q33. Line 586-587. This sentence sounds unclear to me. I believe it must be deleted.

Response: We are sorry for the confusing description of the "However, this embryo division reproductive strategy lacks the clear nurturing and reproductive division of labor seen in social insects, which suggests that C. nacoleiae may also exhibit this evolutionary strategy." We have deleted this sentence.

Q34. Line 598. Add " (Encyrtidae) ".

Response: Done.

Q35. Line 600. Please give "Litomastix" in italics.

Response: Done.

Q36. Line 604. I do not recognize this species. Please spell it out and add names of the author and family.

Response: I' m sorry that my carelessness led to a bad reading experience. We have changed "A. argentata" to "Aenasius arizonensis".

Q37. Line 607. Add " Pang et Wang (Mymaridae)".

Response: Done.

Q38. Line 609. Add " Novković et Kimura (Figitidae) ".

Response: Done.

Q39. Refeerence [44]. change "Biological" to "Biology".

Response: Done.

We would like to express our appreciation to you and reviews on our manuscript and are looking forward to your reply. We hope that the revision can meet the requirements of the journal for review.

Best regards,

Huili Ouyang, Xiaoyun Wang

Feb 9th, 2025

Round 3

Reviewer 1 Report

Comments and Suggestions for Authors

The last version of the manuscript  insects-3419964 improved but there are still important changes to do in order to be approved. Please reed carefully my suggestions in the notes in the attached PDF file. There are 3 aspects that are still missing (or that -at least- should be improved) in the manuscript:

1) the scope of the work: in the last part of the introduction, some questions/hypotheses should be highlighted as the "targets" on which Authors are focusing to  implement their work;

2) the above aspects (questions/hypotheses) should receive an answer during the Discussion and/or Conclusions chapters;

3) Statistical analyses should be improved: in particular the Graph 6 C (describing interesting aspects) is scientifically not suitable and it is not supported by strong statistical analyses.

Comments on the Quality of English Language

The English must be improved (I made some suggestions, but maybe there are still some missing points)

Author Response

Dear editors and reviewers,

We thank the reviewer for the kind consideration and constructive comments on our manuscript entitled “Reproductive Behavior of the Polyembryonic Parasitoid Copidosomopsis nacoleiae (Eady) at different ages” (ID: insects-3419964). We have carefully revised the manuscript and provided the point-by-point response below. The change in the revised manuscript have been highlighted in green for this round. We hope these changes will strengthen our manuscript.

Response to the comments for Reviewer #1

Dear reviewer #1,

We would like to thank you for your professional review work, constructive comments, and valuable suggestions on our manuscript. According to your nice suggestions, we have made extensive corrections to our previous draft, the detailed corrections are listed below.

Q1. Line 56: Delete "include that of Grbić et al".

Response: We gratefully appreciate for your valuable suggestion. We have deleted this part according to the reviewer 's suggestion.

Q2. Line 155-161: This part should be improved (it is too general, without clear targets): to start a research activity, Authors should define some questions, doubts, missing data to find… This is what I am considering the missing part in the introduction.

Response: Thank you for your valuable feedback. We understand the need for clearer definitions of the research questions and gaps in knowledge. In response, we will revise the introduction to explicitly outline the specific questions and uncertainties that our research aims to address. Specifically, we will highlight the key areas where data is currently lacking, such as the developmental process, behavioral patterns, and reproductive rhythms of C. nacoleiae parasitic on D. angustalis. This will provide a clearer context for the study's objectives and its contribution to the field.

Q3. Line 215. To avoid confusion, I am suggesting to mark this subtitle as 2.3..1. Adults (and the next 2.3.2. Larvae; and 2.3.3. Pupae).

Response: Thank you for your remainder. We have modified this part as suggested by the reviewer.

Q4. Line 228: Change "A single newly emerged male and female wasps were paired in a 5 mL" to "Newly emerged male and female wasps were singularly paired into a 5ml…".

Response: We modified the sentence according to the reviewer 's suggestion.

Q5. Lines 246-255: This part must be re-written for several reasons: the division in subchapters must be re-organized; it is not well written. (2) Larvae Parasitized D. angustalis larvae were dissected using a high-depth imaging system (WHICH ONE? PROVIDE MODEL), the number of C. nacoleiae wasp larvae was recorded and physiological and behavioral observations on the pre-imaginal life cycle of the parasitoid were carried out (PLEASE PROVIDE ADDITIONAL INFORMATION/ANTICIPATIONS ON WHAT YOU WERE THINKING TO FIND/DETECT).

Response: We have re-written the materials and methods according to the results. We changed the original 2.3.Life Cycle of C. nacoleiae to 2.2, and changed the original 2.2.Life Cycle and developmental stages of C. nacoleiae to 2.3.Emergence and Longevity of adult C. nacoleiae at various life stages.

Q6. Lines 258: Change "Male and female C. nacoleiae wasps of the same age" to "Males and females of C. nacoleiae wasp of the same age ...".

Response: We have carefully considered all comments from the reviewer and revised our manuscript Accordingly. The manuscript has also been double-checked, and the typos and grammar errors we found have been corrected.

Q7. Lines 261: Supplement the type and model information of the device.

Response: Done. Similar problems in the following text are also supplemented accordingly.

Q8. Lines 284: Change "of Female ..." to "of the female of ...".

Response: Done.

Q9. Lines 303-306: The description of the statistical analyses is too short and there are not bibliographic records.

Response: We appreciate the importance of providing a more detailed description of the statistical analyses. In the revised manuscript, we have expand the different experimental statistical methods section to clearly describe the specific tests and procedures used. Additionally, we will include appropriate bibliographic references to support the chosen statistical approaches.

Q10. Lines 310: At the beginning of a sentence, please write the full genus name.

Response: Thanks for your careful checks. We have changed "C. nacoleiae" to "Copidosomopsis nacoleiae".

Q11. Lines 338: Delete "Only 0.82% of males emerge at 19:00, and no females are observed to emerge in the afternoon".

Response: Done.

Q12. Lines 491: Change "Female C. nacoleiae" to "C. nacoleiae female" .

Response: Done.

Q13. Lines 502: Change "The peak oviposition" to "The peak of the oviposition".

Response: Done.

Q14. Lines 506-515: The statistical analyses for these important parameters are inaccurate: in particular the Graph 6C has some important mistakes.

Response: Thank you for your insightful comment. We appreciate your careful review of the statistical analyses and the identification of potential issues in Graph 6C. Upon re-examination, we have revised the analysis to ensure the accuracy of the statistical methods used. Specifically, we have corrected the incorrect description of the times of oviposition in the text and split Fig.6C into four figures. The updated graphics and results are now presented in a clear and accurate manner in the revised manuscript.

Q15. Lines 517: Change "analysis of oviposition percentages of female C. nacoleiae of different age" to "analysis of C.nacoleiae female oviposition rate at different ages.".

Response: Done.

Q16. Lines 518-519: "Different lowercase letters indicate significant differences ... adults at different time (P < 0.05)" this part should be moved before the Point (C)."Characteristics" is not the correct word; the main problem is that the graph C is not acceptable in this form. It is providing important data, but it is not correct to combine different parameters (such as Times of oviposition, Duration of oviposition, Fec aunditynd Parasitism rate in the same graph, using the unsuitable legenda "Statistical values". Moreover, the Statistical analyses for the parameters presented in the Fig 6C are not reported in the text above.

Response: Thank you for carefully reviewing my paper. Your correction have made me aware of the shortcomings in my research and provided direction for my further improvement. We split Fig.6C into four graphs, and modify the corresponding position in the text.

Q17. Lines 518: "Characteristics" is not the correct word; the main problem is that the graph C is not acceptable in this form. It is providing important data, but it is not correct to combine different parameters (such as Times of oviposition, Duration of oviposition, Fec aunditynd Parasitism rate in the same graph, using the unsuitable legenda "Statistical values". Moreover, the Statistical analyses for the  parameters presented in the Fig 6C are not reported in the text above.

Response: We agree with the reviewer’s suggestion and will incorporate the recommended changes into the manuscript. We split Fig. 6C into four graphs: (C) Times of oviposition Characteristics of oviposition of female C. nacoleiae of different age.  (D) Duration of oviposition of female C. nacoleiae of different age. (E) Fecundity of female C. nacoleiae of different age. (F) Parasitism percentages of female C. nacoleiae of different age. * indicated that there were significant differences in females at different ages (P < 0.05). ns indi-cates that there is no significant difference in females of different ages.

Q18. Lines 589: Change "Aenasius arizonensis" to "A. arizonensis".

Response: Done.

Q19. Lines 622: Change "before 10:00" to "early in the morning (before 10:00 am) during the egg-....".

Response: Done.

We sincerely thanks you for your feedback which would help to improve the quality of our manuscript.

Best regards,

Huili Ouyang, Xiaoyun Wang

Feb 18th, 2025

Round 4

Reviewer 1 Report

Comments and Suggestions for Authors

The manuscript could be accepted in the present form